# The Prognostic Value of Left Atrial Function in Patients with Acute Myocardial Infarction

**DOI:** 10.3390/diagnostics14182027

**Published:** 2024-09-13

**Authors:** Jorun Tangen, Thuy Mi Nguyen, Daniela Melichova, Lars Gunnar Klaeboe, Marianne Forsa, Kristoffer Andresen, Adrien Al Wazzan, Oyvind Lie, Kristina Haugaa, Helge Skulstad, Harald Brunvand, Thor Edvardsen

**Affiliations:** 1ProCardio Center for Innovation, Department of Cardiology, Oslo University Hospital, Rikshospitalet, P.O. Box 4950 Nydalen, 0424 Oslo, Norway; jor-tang@online.no (J.T.); nguyenthumi@gmail.com (T.M.N.); daniela.melichova@sshf.no (D.M.); lgklaeboe@yahoo.no (L.G.K.); m.i.forsa@studmed.uio.no (M.F.); b32542@ous-hf.no (K.A.); adriw@hotmail.fr (A.A.W.); oyvlie@gmail.com (O.L.); k.i.h.h.haugaa@medisin.uio.no (K.H.); helgesku@online.no (H.S.); harbrun@online.no (H.B.); 2Institute for Clinical Medicine, University of Oslo, Sognsvannsveien 9, 0373 Oslo, Norway; 3Department of Cardiology, Hospital of Southern of Norway, Sykehusveien 1, 4838 Arendal, Norway

**Keywords:** acute myocardial infarction, echocardiography, left atrial strain, atrial fibrillation

## Abstract

The prognostic value of left atrial (LA) volume is well-established in acute myocardial infarction (AMI) patients. LA strain provides further patophysological insights. In the present study, we evaluated LA volume and LA strain in AMI patients including those with atrial fibrillation (AF). The aim of the study was to determine if LA strain provide additional prognostic value. Patients with AMI underwent two-dimensional echocardiography within 72 h of admission. The primary outcome was a composite of all-cause mortality and major adverse cardiovascular events. Cox regression analyses were performed. We included 501 patients and during follow-up, 132 patients (26.4%) met the primary outcome. Left ventricular (LV) global longitudinal strain (GLS) (HR 0.94 [95% CI 0.88–0.99], *p* = 0.029), indexed LA volume (LAVi) (HR 1.02 [95% CI 1.00–1.04], *p* = 0.015), and LA reservoir strain (HR 0.96 [95% CI 0.93–0.99], *p* = 0.017) were all independently associated with the primary outcome. A univariate Cox model conducted on the AF patients (*n* = 32) revealed that LA reservoir strain remained significantly associated with the primary outcome, while LV GLS and LAVi were not significant. The prognostic value of LA reservoir strain was comparable to LA volume and LV GLS, and might even be better in AF patients.

## 1. Introduction

An enlarged left atrial (LA) volume has been widely acknowledged as a strong predictor of adverse outcomes following acute myocardial infarction (AMI) [1,2]. The assessment of LA strain using speckle tracking echocardiography (STE) has garnered attention as a method to detect early changes in atrial function [3,4]. A study conducted in 2012 highlighted that LA deformation analysis was a more sensitive parameter, providing independent and supplementary prognostic information compared to other conventional LA measurements [5]. LA strain comprises three components (reservoir, conduit, and pump) in relation to atrial functional physiology [3,6].

The function of the LA is closely linked to the function of left ventricle (LV), and together they play a key role in maintaining optimal cardiac performance. The LA affects the filling of the LV through its reservoir, conduit, and pump functions, while the LV function affects the atrium throughout the cardiac cycle [7]. LA remodeling is connected to LV remodeling, and function of the LA is crucial in sustaining optimal cardiac output even in the presence of impaired LV relaxation and reduced LV compliance [8]. In patients with myocardial infarction (MI), LV stroke volume is relatively maintained despite the impaired LV function. In these patients, the LA effectively compensates by working harder to transport blood to the left ventricle during LV diastole. As the LA diameter increases, its output also increases, contributing to the maintenance of a normal stroke volume. This function of the LA can be attributed to the Frank–Starling mechanism [8]. Additionally, the contractile function of the LA may decrease in the presence of severe dilation, exceeding the optimal Frank–Starling relationship. Therefore, LA function can provide additional valuable information, beyond LA volume measurements [7].

Atrial fibrillation (AF) is considered a progressive condition that typically initiates from elevated hemodynamic load in the heart and/or changes in the structure of the atria, such as the expansion of chambers and the development of interstitial fibrosis [9]. Gaining a better comprehension of the structure and function of the atria in AF patients has the potential to enhance our predictive capabilities for adverse outcomes.

The aim of this study was to determine whether strain echocardiography measurements of LA function provide additional prognostic value in a diverse population of patients following AMI, including those with AF.

## 2. Materials and Methods 

### 2.1. Population of the Study 

We enrolled patients from 2014 to 2022 who had suffered from AMI in a prospective multicenter cohort study conducted at three hospitals in Norway: The Hospital of Southern Norway in Arendal and Kristiansand, as well as Oslo University Hospital, Rikshospitalet. This study was conducted as part of the IMPROVE study (clinical trials: NCT02286908), a prospective, observational, multicenter study which aimed to investigate the predictive value of strain echocardiography in assessing the risk of patients with heart diseases. 

AMI was confirmed based on a combination of criteria that meet the established guidelines [10,11]. In our study, patients with type 2 infarction and other causes of chest pain were excluded. 

A total of 517 AMI patients were evaluated for the present study. We excluded 16 patients (3.1%) due to bad image quality, left bundle branch block, right ventricular pacing, severe mitral regurgitation (MR), or mitral stenosis (Figure 1) [6]. Hence, the final study population consisted of 501 patients. Patients were examined with electrocardiography for AF at admission. 

The median follow-up period was 4.3 years (interquartile range (IQR) 3.2 to 5.0 years). Informed consent was obtained from all patients, and the study was approved by the local ethics committee (REK 2013/573). The study was carried out in accordance with the ethical principles outlined in the Declaration of Helsinki. Treatment of study patients followed current guidelines [10,11,12], and the study did not interfere with acute treatment. Patients who experienced in-hospital death were excluded from the study. 

Baseline demographic data and presenting features were documented upon admission (Table 1). Clinical signs of heart failure (HF) at admission were assessed using the New York Heart Association Functional Classification. Upon admission, high-sensitivity cardiac troponin I or T levels were measured. Levels above the 99th percentile of the upper reference limit (>45 ng/L or >14 ng/L, respectively) were considered elevated. Medications prescribed at the time of discharge were documented. Patients were monitored until the end of the study or until death. 

### 2.2. Outcome Variables

The primary outcome of the study was a composite of all-cause mortality and major adverse cardiovascular events (MACE). MACE encompassed hospitalizations due to MI, HF, stroke, and/or cardiac arrest. 

### 2.3. Echocardiography 

All study patients underwent a comprehensive two-dimensional transthoracic echocardiography (Vivid E9/E95 (GE, Vingmed, Horten, Norway)) within 72 h after admission and invasive procedures. The echocardiograms were performed by experienced cardiologists/sonographers and were reviewed by cardiologists with advanced training in echocardiography. The data were digitally recorded for analysis using EchoPac version 204 software from GE Healthcare, Vingmed. 

LV volume and ejection fraction (EF) were measured using the biplane modified Simpson´s method, with the ranges and severity of cutoff values for LV EF aligned with the guidelines [13].

LV global longitudinal strain (GLS) was calculated as the average of segmental peak systolic myocardial strain from the three standard apical views using two-dimensional STE [14]. All GLS calculations in this study were reported as absolute values. 

LA volume was assessed using the biplane area-length method from the apical 4- and 2-chamber views. Measurements were taken at end systole and indexed for body surface area [15]. LA dilatation was defined as an indexed LA volume (LAVi) greater than 34 mL/m^2^ [13,16]. LA volume was used as a continuous variable in the univariate and the multivariate cox regression analysis. 

### 2.4. Left Atrial Function 

LA strain encompasses three components of atrial functional physiology. Reservoir strain represents atrial function during systole, occurring before mitral valve opening, indicating atrial filling. Subsequently, the conduit phase corresponds to the passive flow from the left atrium to the LV in early diastole. Lastly, contraction strain or pump strain occurs during the booster phase of atrial systole (Figure 2) [3,6].

LA strain was analyzed using LA automated function imaging package in the software and was based on LA-focused and non-foreshortened four-chamber views [6]. The R wave was used as reference, and zero strain was set at the R wave [6,17]. The pre-atrial contraction timing was manually adjusted when needed, based on identification of the P wave in the electrocardiogram and the sudden fall in the strain curve associated with atrial contraction. We used LA strain as a continuous variable in the univariate and the multivariate cox regression analysis. 

LV filling pressure was evaluated according to the 2022 European Association of Cardiovascular Imaging consensus document [15,18]. This algorithm integrated five echocardiographic parameters (E/A, E/é, tricuspid regurgitation velocity, LAVi, and LA reservoir strain), each with defined cutoff values. LA reservoir strain was the latest addition to this algorithm and was utilized when the other criteria could not definitely determine whether the filling pressure was elevated or not. The LA reservoir strain was treated as a binary variable in this algorithm. 

To assess LV diastolic function, pulsed-wave Doppler of the mitral valve inflow was obtained by placing the Doppler sample volume between the tips of the mitral leaflets. The early (E) and late (A) peak diastolic velocities and E-wave deceleration time were measured. E/é ratio was obtained by dividing E by é, which was measured using color-coded tissue Doppler imaging at the septal side of the mitral annulus in the apical 4-chamber view [16].

Mitral regurgitation (MR) was classified as mild in cases where there was no MR jet or a small central jet area accounting for less than 20% of the LA on Doppler, and the vena contracta measured less than 0.3 cm. MR was categorized as moderate when the central jet accounted for 20–40% of the LA and the vena contracta measured less than 0.7 cm [19]. We dichotomized MR in absent/mild or moderate regurgitation.

### 2.5. Coronary Artery Lesions and Treatment

Coronary lesions were evaluated using visual assessment of angiographic stenosis, with a threshold of >50% diameter reduction considered significant. Patients were treated with primary percutaneous coronary intervention (PCI) and/or optimal medical therapy. 

### 2.6. Statistical Analysis

Continuous data were presented as mean with standard deviation or median with interquartile range and compared with Student´s *t*-test or Mann–Whitney U test, respectively. Categorical data were presented as number with percentages and comparing using chi-squared or Fisher’s exact tests, as appropriate. Statistical significance was determined using a two-sided *p*-value of < 0.05.

Cox proportional hazard analyses were conducted to assess the association between LA volume and LA strain as well as other variables potentially influencing mortality and survival. Carefully selected echocardiographic variables significantly associated with primary outcome in the univariable models (Table 2) were entered into the final multivariable model (Table 3) together with age. We examined the correlation between the variables that were intended to be included in the selected model. With a moderate to high degree of correlation (>50%) [20], the variables were not used in the same model or removed. To examine the independent and added prognostic value of LA volume and LA strain, we stepwise added these variables to the multivariate Cox model consisting of age, LV filling pressure, moderate MR, and LV GLS (Table 3). LV EF was excluded from the multivariate Cox model because of significant correlation with LV GLS (r = 0.60, *p* < 0.001) [15]. We analyzed LA reservoir strain and LA conduit and pump strain in two different models (Model 3a and 3b) due to high correlation between LA reservoir and conduit strain (r = 0.68, *p* < 0.001) and LA reservoir and pump strain (r = 0.65, *p* < 0.001). Akaike Information Criterion (AIC) and concordance index (Harrell´s C statistic) were calculated at each step in Model 1 to Model 3 to test the strength and quality of the models.

We performed univariate Cox regression analysis in patients with AF (*n* = 32) due to limited sample size. 

We used Kaplan–Meier survival curves to plot survival free from all-cause mortality and MACE. Differences between the curves were assessed by the log-rank test (Figure 3).

Finally, intra- and interobserver variability were determined by repeating the LA measurements from the same dataset in 25 randomly selected patients and expressed by intraclass correlation coefficients. 

Statistical analyses were performed using Stata software version 17.0 (Copyright 1985–2023 StataCorp LLC, StataCorp, 4905 Lakeway Drive, College Station, TX 7785, USA) and IBM, SPSS version 29 (SPSS Inc., Chicago, IL, USA).

## 3. Results 

### 3.1. Patient Characteristics 

Table 1 presents a summary of the clinical and echocardiographic characteristics. The average age of the patients was 69 ± 12 years; 26% were female. Half of the patients had a history of hypertension (HT), and one-fifth had diabetes mellitus (DM). Three-quarters of the patents experienced their first AMI. In 92.6% of the cases, patients were treated by PCI with the left anterior descending artery being the most frequently treated. Less than one-third of the patients exhibited multivessel coronary artery disease, characterized by stenosis in two or more coronary arteries. We included 438 patients (87.4%) with non-ST-elevation myocardial infarction and 63 (12.6%) with ST-elevation myocardial infarction.

### 3.2. Predictors of Adverse Outcome

During follow-up (median 4.3 years (IQR 3.2 to 5.0 years)), 132 patients (36.4%) experienced the primary outcome. Among these, a total of 53 patients (10.6%) died. Additionally, readmissions to the hospital during the follow-up period included 52 (10.4%) with MI, 22 (4.4%) with stroke, 33 (6.6%) with HF, and 11 (2.2%) with cardiac arrest.

In Table 3, the multivariate analysis (Model 3a) demonstrated that LA reservoir strain was significantly associated with and was an excellent predictor of all-cause mortality and MACE. LV GLS and LAVi were also significant predictors of primary outcome. However, LA conduit and pump strain were not significant predictors of primary outcome in our study (Model 3b). 

The AIC indicated that Model 3 was best fit for the data in Table 3, and the Harrell´s C statistic confirmed that Model 3a had good probability of experiencing the primary outcome. Model 2 and Model 3a yielded remarkable similar predictions of outcome events, with Harrell´s C statistics of 0.705 and 0.707, respectively. Comparing these two models, there was no significant difference between their performance (*p* = 0.965). 

In a separate analysis in patients with AF (*n* = 32), a univariate Cox regression analysis revealed that LA reservoir strain was a significant predictor of the primary outcome (HR 0.89 [95% CI 0.82–0.96], *p* = 0.004). LAVi was only borderline significant in patients with AF (HR 1.03 [1.00–1.06], *p* = 0.052). Patients with AF (*n* = 32) had a significant higher risk of primary outcome compared to those without AF (*n* = 469), *p* < 0.001. 

LA reservoir strain < 16.3% identified patients at higher risk for mortality, which is illustrated in Table 1. Patients with LA reservoir strain < 16.3% exhibited poorer survival free from all-cause mortality and/or readmission to hospital compared to those with LA reservoir strain ≥ 16.3% (*p* < 0.001), as demonstrated with Kaplan–Meier curves (Figure 3) [21]. 

### 3.3. The Impact of Atrial Fibrillation and Left Ventricular Filling Pressure 

Patients with AF had a significant reduction in LV GLS (11.5 ± 4.3% vs. 15.1 ± 3.2%, *p* < 0.001), an increased LAVi (43.1 ± 13.9 mL/m^2^ vs. 29.6 ± 10.2 mL/m^2^, *p* < 0.001), and a decrease in LA reservoir strain (13.1 ± 7.1% vs. 25.9 ± 7.8% *p* < 0.001) compared to those without AF. 

Figure 4 shows the negative impact of LV filling pressure on LA function (reservoir-, conduit-, and pump strain), using Violin plots.

Male patients were found to have a higher frequency of normal to slightly reduced LA reservoir strain compared to female patients (25.7 ± 8.4% vs. 23.3 ± 8.0% [*p* = 0.001], respectively). Patients with decreased LA reservoir strain were observed to be older than those with normal LA reservoir strain (76 ± 11 years vs. 68 ± 12 years [*p* < 0.001], respectively). 

Inter- and intraobserver variability repeating the LA volume measurements were excellent with intraclass correlation coefficients at 0.986 and 0.984, respectively (both *p* < 0.001). Inter- and intraobserver variability repeating the LA reservoir/conduit/pump strain were excellent to very good with intraclass correlation coefficients at 0.950/0.931/0.755 and 0.821/0.779/0.824 (*p* < 0.001 for all three LA strain variables).

## 4. Discussion 

This is the first study to examine the prognostic value of LA strain in a diverse population, encompassing patients diagnosed with either known or newly developed AF shortly after experiencing AMI.

The key discoveries from this prospective evaluation can be outlined as follows: (1) The assessment of LA function using LA reservoir strain offers valuable prognostic insight and enhances the predictive capacity of LV GLS and LA volume for primary clinical outcome following AMI; and (2) LA reservoir strain represents a promising innovative approach to quantify LA function in patients with AF. 

The importance of LA volume as a predictor of post-AMI survival was introduced more than 2 decades ago [1]. Our study sought to elucidate the prognostic significance of LA reservoir strain in patients with AMI. The results from our study indicate that LA reservoir strain exhibits a robust predictive value after AMI, which for patients with AF might even be superior to LV GLS and LA volume. Importantly, our study includes patients with AF, unlike others which have excluded AF patients from their study population [22,23]. 

In later years, the significance of LA strain in predicting post-AMI survival was introduced as a promising measurement [22,23]. However, there are conflicting findings regarding the prognostic value of LA strain after AMI. Antoni et al. introduced LA strain as a predictor of post-AMI survival in 2011. With their retrospective study, they demonstrated that LA strain provided additional prognostic information beyond LA volume for predicting adverse outcome [23]. Ersboel et al. assessed the interrelation of LA reservoir strain and LV GLS in a large-scale study of AMI-patients for prognosis. They did not find any independent effect of LA reservoir strain when adjusted for LV GLS and LA volume [22]. None of those studies included patients with AF. Our study demonstrated that LA reservoir strain provided prognostic information comparable to LA volume. Both variables offered valuable prognostic insights in patients with AMI. We also found that LA reservoir strain may be a more effective tool than LA volume in patients with AF. 

Interestingly, it has been shown that the predictive utility of LA volume for cardiovascular events in AF was poor. However, LA volume was a robust marker in subjects with sinus rhythm [24]. This was in line with our findings where we found significant correlation of LA volume for primary outcome except in patients with AF only, while LA reservoir strain remained a significant marker in AF.

LV remodeling takes place in post-AMI patients, alongside concurrent effects on the LA. The remodeling of the LA commences within the first week following an AMI and subsequently progresses gradually over a period of up to 3 months [23]. The relaxation of the LA, as indicated by its reservoir function, plays an important role during acute ischemia. As a result of the heightened stiffness and increased filling pressures in the LV chamber, there may be an elevation in the pressure within the LA [25,26]. Maintaining sufficient LV filling is needed, and to achieve this, a preserved LA reservoir function is important. This reservoir function allows the LA to withstand the effects of increased pressure. On the other hand, in patients with non-compliant LA and reduced reservoir function, there is a significant impairment in LV filling, thereby increasing the risk of adverse outcomes [23].

The main function of the LA is to regulate the filling of the LV [27]. The ability of the LA to act as a reservoir is determined by its compliance during ventricular contraction, as well as the volume of blood in the LV at the end of the contraction [26]. Patients with HF (irrespective of LV EF) have been known to be in an increased risk of developing high LV filling pressure, resulting in alterations in the loops described above. A recent study explored the relationship between LA strain and LV diastolic dysfunction. In patients with LV diastolic dysfunction with normal LV EF, LA strain presented good correlation with invasive left ventricular end diastolic pressure (LVEDP), and LA reservoir strain was superior to LA conduit and pump strain, and E/é, to predict elevated LVEDP [28]. Our findings align with the notion that LA strain measurements were affected by LV diastolic dysfunction and high LV filling pressure. We discovered a significant impairment in all LA strain measurements (reservoir, conduit, and pump), when LV filling pressure was elevated. This highlighting the importance of acknowledging the interplay between atrial function and ventricular function throughout the cardiac cycle. 

AF is a condition often linked to LA remodeling, involving both structural and functional alterations [7]. One study has shown that LA wall fibrosis, as evaluated using delayed enhancement cardiac magnetic resonance (CMR) imaging, was associated with impaired function, as measured by LA strain, and both factors were linked to an increased risk of AF [7,9]. In 2020, Kim et al. demonstrated that LA reservoir strain provided incremental diagnostic utility compared to LA geometry shown by CMR imaging. LA reservoir strain was particularly effective for stratifying the presence and severity of diastolic dysfunction and enhanced the prediction of AF and HF following myocardial infarction (MI) [29]. This supports our findings that patients with AF had a notable negative effect on LA function (and primary outcome), which leads us to believe that our AF patients exhibited differing degrees of LA wall fibrosis. 

### 4.1. Study Limitations

This study underscored the significance of early echocardiography in evaluating LA volume and LA strain within the first 72 h of admission for patients with AMI. Nonetheless, it is essential to recognize that during this early phase of LV dysfunction, the complete remodeling process following MI has not yet taken place. Consequently, it is important to exercise caution when extrapolating our finding to studies conducted at later stages.

Another consideration relates to the intrinsic three-dimensional nature of the LA. In our study, LA volume and LA strain were measured in a two-dimensional manner. Three-dimensional imaging might offer an even more accurate method of defining LA volume and function. 

Furthermore, the high percentage of revascularized patients (92.6%) is attributed to the inclusion of MI type 1 cases, with type 2 cases excluded. This differentiation is important in understanding outcomes related to revascularization, and to exercise caution when extrapolating our results to larger and more diverse population groups. Another important point is that, since the majority of the study population underwent revascularization, the results cannot be generalized to patients with MI with nonobstructive coronary arteries (MINOCA). 

The number of patients with AF in our study sample was limited. The results from the patients with AF should therefore be interpreted with caution.

### 4.2. Clinical Perspectives

Our findings have two significant clinical implications. Firstly, it may be useful to include LA reservoir strain together with LA volume in the evaluation of patients with AMI, as it proved additional prognostic information on severe outcomes. Secondly, an enlarged LA volume and/or reduced LA reservoir strain can serve as indicators for the need to intensify treatment for conditions as HF with or without impaired LV EF and elevated LV filling pressure, which might have a considerable impact on prognosis. LA reservoir strain appeared to be a promising measurement for predicting adverse outcome in patients with AF. Both, LA volume and LA strain provided important insights on severe outcome after an AMI and should be routinely measured in clinical practice.

## 5. Conclusions 

Our findings demonstrate LA reservoir strain as an independent predictor of all-cause mortality and MACE in patients with AMI. The prognostic value of LA reservoir strain was comparable to LA volume and LV GLS, and might even be superior in AF patients. 

## Figures and Tables

**Figure 1 diagnostics-14-02027-f001:**
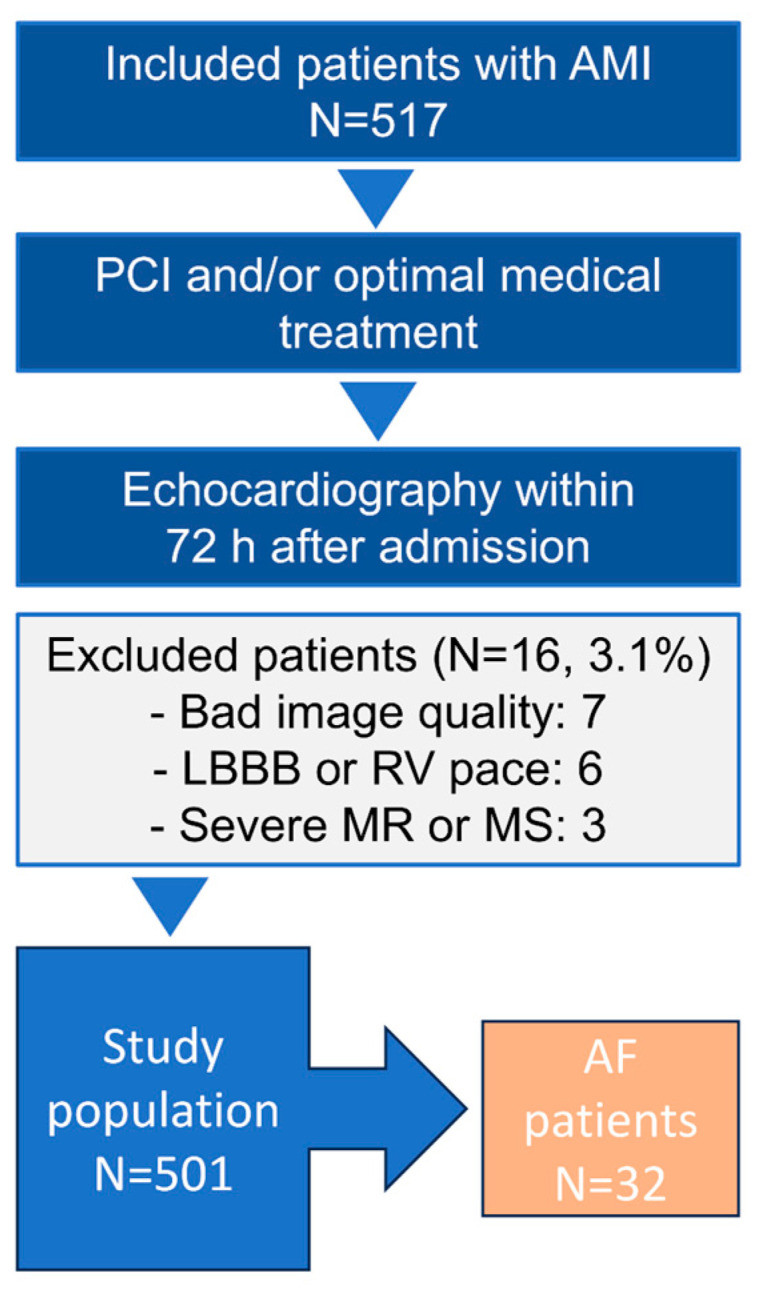
Flow chart of the study population; AMI = acute myocardial infarction; AF = atrial fibrillation; LBBB = left bundle branch block; MS = mitral stenosis; MR = mitral regurgitation; PCI = percutaneous coronary intervention; RV = right ventricle.

**Figure 2 diagnostics-14-02027-f002:**
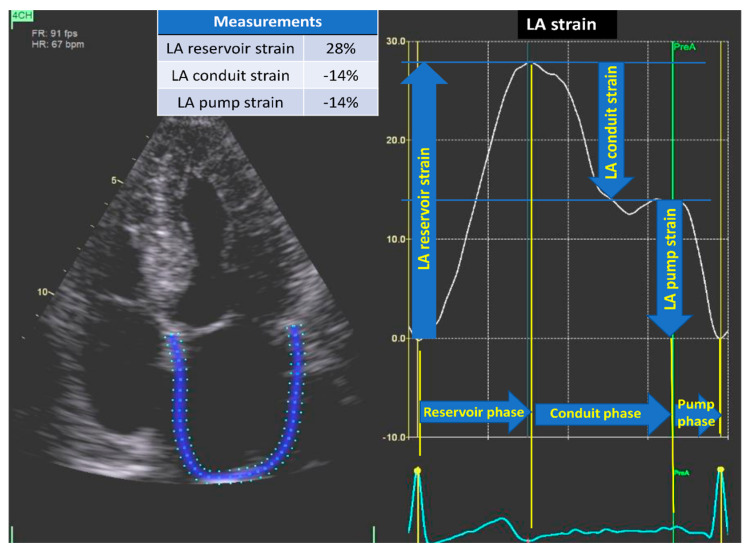
LA strain. Two-dimensional STE from the apical four-chamber view demonstrating LA strain and different phases. LA reservoir strain represents atrial function during systole, occurring before mitral valve opening, indicating atrial filling. LA conduit strain corresponds to the passive flow from the left atrium to the left ventricle in early diastole. LA pump strain occurs during the active phase of atrial systole. LA = left atrial; STE = speckle tracking echocardiography.

**Figure 3 diagnostics-14-02027-f003:**
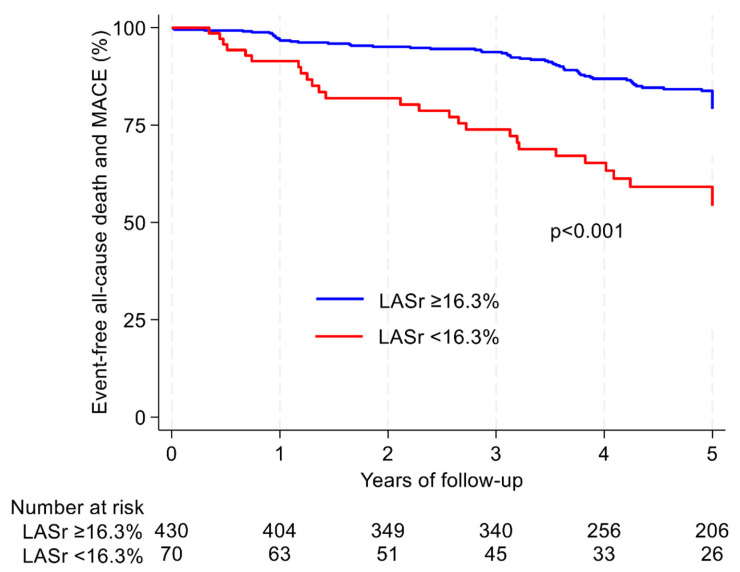
Kaplan–Meier curves indicating the time to onset of all-cause death and MACE according to AMI-patients. The log-rank test was used to compare the outcome distribution between normal (blue curve) and pathological values (red curve) in patients with LASr ≥ 16.3% and <16.3%. Number at risk table gives information about how many patients at risk at a given time for the two groups. AMI = acute myocardial infarction; LASr = left atrial reservoir strain; MACE = major adverse cardiovascular events.

**Figure 4 diagnostics-14-02027-f004:**
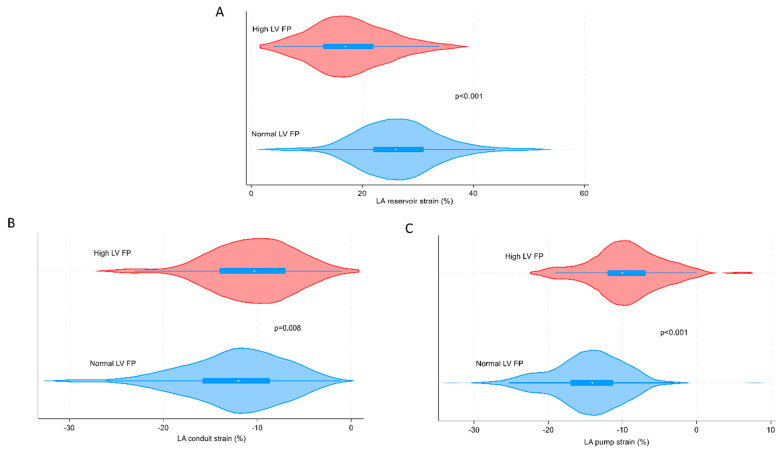
Violin plots indicating LV FP´s impact on LA reservoir strain (**A**), LA conduit strain (**B**), and LA pump strain (**C**). The study population includes 501 patients for the measurements of LA reservoir strain. LA conduit and pump strain are measured in the same study population as LA reservoir strain except from excluding AF patients (*n* = 469). AF = atrial fibrillation; LA = left atrial; LV FP = left ventricular filling pressure.

**Table 1 diagnostics-14-02027-t001:** Baseline Characteristics of Study Cohort.

	Overall (*n* = 501)	Survivors (*n* = 448)	Deceased (*n* = 53)	*p* (Survived vs. Deceased)
Age, years	69 ± 12	68 ± 12	80 ± 10	<0.001
Gender				
Female, *n* (%)	130 (25.9)	117 (26.1)	13 (24.5)	0.804
Male, *n* (%)	371 (74.1)	331 (73.9)	40 (75.5)	0.804
STEMI, *n* (%)	63 (12.6)	55 (12.3)	8 (15.1)	0.559
NSTEMI, *n* (%)	438 (87.4)	393 (87.7)	45 (84.9)	0.559
Risk factors				
Smoke (current and last 10 years), *n* (%)	294 (60.2)	265 (60.9)	29 (54.7)	0.384
Body mass index, kg/m^2^	27.3 ± 4.0	27.4 ± 4.0	26.8 ± 4.3	0.330
Weight, kg				
Female	74.6 ± 13.7	74.6 ± 13.7	71.9 ± 14.2	0.467
Male	88.3 ± 13.9	88.6 ± 13.5	85.5 ± 16.4	0.176
Hypertension, *n* (%)	243 (48.5)	201 (44.9)	42 (79.2)	<0.001
Diabetes Mellitus, *n* (%)	104 (20.8)	88 (19.7)	16 (30.2)	0.076
Hypercholesterolemia, *n* (%)	133 (28.1)	118 (27.8)	15 (30.6)	0.682
Previous HF, *n* (%)	17 (3.4)	11 (2.5)	6 (11.3)	0.001
Previous MI, *n* (%)	112 (22.4)	87 (19.4)	25 (47.2)	<0.001
First time MI, *n* (%)	389 (77.6)	361 (80.6)	28 (52.8)	<0.001
Previous CABG, *n* (%)	28 (5.6)	20 (4.5)	8 (15.1)	0.001
SBP, mmHg	133 ± 20	132 ± 19	140 ± 27	0.009
DBP, mmHg	78 ± 13	79 ± 12	79 ± 13	0.998
Heart rate, bpm	71 ± 13	70 ± 13	75 ± 15	0.013
Clinical assessment				
NYHA ≥ 2 admission, *n* (%)	88 (17.6)	66 (14.7)	22 (41.5)	<0.001
Echocardiography				
LV EF, %	54 ± 9	55 ± 9	50 ± 11	<0.001
LV GLS, %	15.0 ± 3.3	15.2 ± 3.2	13.1 ± 4.1	<0.001
LAVi, mL/m^2^	30.4 ± 11.0	29.3 ± 10.2	39.6 ± 13.2	<0.001
LA reservoir strain, %	25.0 (20.0 to 30.0)	25.9 (25.1 to 26.1)	18.6 (16.3 to 20.9)	<0.001
LA conduit strain *, %	−11.7 (−15.2 to −8.0)	−12.0 (−15.8 to −9.0)	−10.3 (−14.0 to −7.0)	0.002
LA pump strain *, %	−13.8 (−17.0 to −10.0)	−14.0 (−17.2 to −11.0)	−12.9 (−16.0 to −8.9)	<0.001
E/A ratio	0.93 ± 0.36	0.94 ± 0.35	0.91 ± 0.45	0.782
Septal é velocity, cm/s	7.0 ± 2.0	6.6 ± 1.9	5.5 ± 1.6	<0.001
E/é ratio	11.2 ± 4.9	10.7 ± 4.0	15.3 ± 8.6	<0.001
Elevated LV filling pressure, *n* (%)	85 (17.0)	63 (14.1)	22 (41.5)	<0.001
MR grade 2, *n* (%)	23 (5.0)	17 (3.8)	8 (15.4)	<0.001
TR, *n* (%)	247 (68.4)	215 (67.6)	27 (75.0)	0.366
Blood samples				
Hs-cTnI max, ng/L	1009 (223 to 3338)	3113 (1878 to 4348)	8878 (519 to 18,275)	0.012
Hs-cTnT max, ng/L	568 (165 to 2070)	1946 (1571 to 2321)	2174 (1213 to 3135)	0.695
Creatinin, µmol/L	78 (69 to 91)	82 (78 to 86)	110 (92 to 128)	<0.001
NT-proBNP, ng/L	509 (178 to 1559)	1289 (949 to 1630)	4160 (1793 to 6528)	<0.001
Treatment				
Primary reperfusion/PCI, *n* (%)	464 (92.6)	418 (93.3)	46 (86.8)	0.087
Coronary artery with significant stenosis				
Left main coronary artery, *n* (%)	50 (10.0)	37 (8.3)	13 (24.5)	<0.001
Left anterior coronary artery, *n* (%)	255 (50.9)	231 (51.6)	24 (45.3)	0.387
Left circumflex coronary artery, *n* (%)	204 (40.7)	178 (39.7)	26 (49.1)	0.191
Right coronary artery, *n* (%)	171 (34.1)	157 (35.0)	14 (26.4)	0.210
Multivessel disease, *n* (%)	154 (30.7)	131 (29.2)	23 (43.4)	0.035

Data are expressed as numbers (%), mean ± SD or median (IQR) as appropriate. * Study population without AF (*n* = 469). AF = atrial fibrillation; Bpm = beats per minute; CABG = coronary artery bypass grafting; DBP = diastolic blood pressure; EF = ejection fraction; GLS = global longitudinal strain; HF = heart failure; HR = hazard ratio; Hs-cTnI = high-sensitivity cardiac troponin I; Hs-cTnT = high-sensitivity cardiac troponin T; IQR = interquartile range; LA = left atrial; LAVi = indexed left atrial volume; LV = left ventricular; Max = maximum; MI = myocardial infarction; MR = mitral regurgitation; NSTEMI = non-ST-segment Elevation Myocardial Infarction; NT-proBNP = N-terminal pro-B-type natriuretic peptide; NYHA = New York Heart Association; PCI = percutaneous coronary intervention; SBP = systolic blood pressure; STEMI = ST-segment Elevation Myocardial Infarction; TR = tricuspid regurgitation.

**Table 2 diagnostics-14-02027-t002:** Predictors of All-cause Mortality and MACE by Univariable Cox Regression *.

Covariate	HR (95% CI)	*p*
Clinical information		
NYHA ≥ 2 admission	1.92 (1.29–2.85)	0.001
Risk factors		
Age, years	1.03 (1.01–1.04)	<0.001
Hypertension	1.83 (1.29–2.60)	0.001
Diabetes	1.70 (1.16–2.49)	0.007
History of MI	2.76 (1.94–3.93)	<0.001
History of CABG	2.50 (1.43–4.34)	0.001
History of HF	2.57 (1.39–4.77)	0.003
First MI	0.36 (0.25–0.52)	<0.001
SBP, mmHg	1.01 (1.00–1.02)	0.008
Heart rate, bpm	1.02 (1.00–1.03)	0.008
Echocardiography		
LV function		
LV EF, %	0.97 (0.95–0.98)	<0.001
LV GLS, %	0.87 (0.83–0.92)	<0.001
E/é ratio	1.11 (1.09–1.14)	<0.001
Septal é velocity, cm/s	0.77 (0.69–0.86)	<0.001
LV filling pressure	2.62 (1.83–3.76)	<0.001
Moderate MR	3.46 (1.56–7.68)	0.002
TR	2.30 (1.37–3.85)	<0.001
LA volume and function		
LAVi, mL/m^2^	1.04 (1.03–1.06)	<0.001
LA reservoir strain, %	0.92 (0.90–0.94)	<0.001
LA conduit strain **, %	1.07 (1.03–1.11)	<0.001
LA pump strain **, %	1.06 (1.03–1.10)	<0.001
Blood samples		
Hs-cTnI max, per 100 ng/L	1.01 (1.00–1.02)	0.036
Hs-cTnT max, per 100 ng/L	1.00 (1.00–1.01)	0.570
Creatinine, per 100 µmol/L	1.23 (1.03–1.48)	0.024
NT-proBNP, per 100 ng/L	1.01 (1.00–1.01)	<0.001
Infarct treatment		
Primary reperfusion/PCI	0.42 (0.25–0.68)	0.001

* All-cause mortality and MACE available for 501 patients. ** All-cause mortality and MACE available for 469 patients without AF. AF = atrial fibrillation; Bpm = beats per minute; CABG = coronary artery bypass grafting; EF = ejection fraction; GLS = global longitudinal strain; HF = heart failure; HR = hazard ratio; Hs-cTnI = high-sensitivity cardiac troponin I; Hs-cTnT = high-sensitive cardiac troponin T; LA = left atrial; LAVi = indexed left atrial volume; LV = left ventricular; MACE = major adverse cardiovascular events; MI = myocardial infarction; MR = mitral regurgitation; NYHA = New York Heart Association; NT-proBNP = N-terminal pro-B-type natriuretic peptide; PCI = percutaneous coronary intervention; SBP = systolic blood pressure; TR = tricuspid regurgitation.

**Table 3 diagnostics-14-02027-t003:** Stepwise Multivariate Cox Regression Models for Predictors of All-cause Mortality and MACE *.

	Model 1 (Age, LV FP, moderate MR and LV GLS)		Model 2 (Model 1 + LAVi)		Model 3a (Model 2 + LASr)		Model 3b (Model 2 + LASc and LASp)	
Covariate	HR (95% CI)	*p*	HR (95% CI)	*p*	HR (95% CI)	*p*	HR (95% CI)	*p*
Age	1.02 (1.01–1.04)	0.007	1.02 (1.00–1.03)	0.041	1.01 (1.00–1.03)	0.117	1.01 (1.00–1.03)	0.144
LV FP	1.74 (1.15–2.63)	0.008	1.28 (0.81–2.03)	0.285	1.15 (0.72–1.82)	0.561	1.47 (0.91–2.39)	0.116
Moderate MR	1.33 (0.75–2.36)	0.333	1.30 (0.73–2.31)	0.370	1.15 (0.64–2.06)	0.634	1.07 (0.55–2.07)	0.840
LV GLS	0.90 (0.85–0.95)	<0.001	0.91 (0.86–0.96)	0.001	0.94 (0.88–0.99)	0.029	0.93 (0.87–0.99)	0.028
LAVi			1.03 (1.01–1.04)	0.001	1.02 (1.00–1.04)	0.015	1.02 (1.00–1.04)	0.033
LASr					0.96 (0.93–0.99)	0.017		
LASc **							1.04 (0.99–1.08)	0.093
LASp **							1.00 (0.96–1.05)	0.820
AIC	1391		1382		1379		1261	
Harrell´s C	0.678		0.705		0.707		0.696	

* All-cause mortality and MACE available for 501 patients. Includes three models with expanding number of covariates. Model 3 is divided in two (a and b). ** All-cause mortality and MACE available for 469 patients without AF. AIC = Akaike Information Criterion; GLS = global longitudinal strain; Harrell´s C = concordance statistic; HR = hazard ratio; LA = left atrial; LAVi = indexed left atrial volume; LASc = left atrial conduit strain; LASp = left atrial pump strain; LASr = left atrial reservoir strain; LV = left ventricular; MACE = major adverse cardiovascular events; MR = mitral regurgitation.

## Data Availability

All relevant data are presented in figures or tables within the paper.

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
