# Peer review of "The Prognostic Value of Left Atrial Function in Patients with Acute Myocardial Infarction"

_diagnostics, 2024, doi:10.3390/diagnostics14182027_

Round 1
Reviewer 1 Report
Comments and Suggestions for Authors
The main objective of the work under consideration was to evaluate whether LA strain affects outcomes in patients with acute myocardial infarction in general and in the group of patients with atrial fibrillation. The topic of the article is relevant and original. The results obtained by the authors fill the gap in this issue. The authors were the first to show that LA reservoir strain is an independent predictor of all-cause mortality and MACE in patients with acute myocardial infarction. Evaluation of LA function using LA reservoir strain increases the prognostic significance of LV GLS and LA volume in predicting outcomes in patients with acute myocardial infarction. The authors' conclusions are justified, they follow from the results and meet the purpose of the work. There are no comments on the methodology of the work and statistical processing. All references provided are relevant. There are too many tables in the analyzed article. The authors are advised to shorten the tables if they deem it possible.
Reviewer 2 Report
Comments and Suggestions for Authors
The authors aimed to evaluate if strain echo measurement of left atrium function can provide additional prognostic value in patients following MI, including those with Afib.
I congratulate the authors on choosing this important topic. Please find my comments below.
The abstract and introduction are concise and well written.
The methodology is appropriate and results are well described.
The discussion section is well written with limitations addressed. Most of the study population had revascularization and these results cannot be extrapolated to MINOCA (MI with nonobstructive coronaries). Please add this to the limitations section.
